# Hydrolyzable Additive-Based Silicone Elastomers: A New Approach for Antifouling Coatings

**DOI:** 10.3390/polym11020305

**Published:** 2019-02-12

**Authors:** Laure Gevaux, Marlène Lejars, André Margaillan, Jean-François Briand, Robert Bunet, Christine Bressy

**Affiliations:** 1Laboratoire Matériaux Polymères Interfaces Environnement Marin (MAPIEM), Université de Toulon, EA 4323, 83957 La Garde, France; marlene.lejars@univ-tln.fr (M.L.); andre.margaillan@univ-tln.fr (A.M.); jean-francois.briand@univ-tln.fr (J.-F.B.); christine.bressy@univ-tln.fr (C.B.); 2Institut Océanographique Paul Ricard, Ile des Embiez, 83140 Six-Fours-les-Plages, France; robert.bunet@institut-paul-ricard.org

**Keywords:** silicone elastomer, hydrolyzable polymers, amphiphilic polymers, surface chemistry, Fouling Release Coating

## Abstract

Fouling Release Coatings are marine antifouling coatings based on silicone elastomers. Contrary to commonly used biocide-based antifouling coatings, they do not release biocides into the marine environment, however, they suffer from poor antifouling efficacy during idle periods. To improve their antifouling performances in static conditions, various amounts of hydrolyzable polymers were incorporated within a silicone matrix. These hydrolyzable polymers were chosen for the well-known hydrolytic degradation mechanism of their main chain, e.g. poly(ε-caprolactone) (PCL), or of their ester pending groups, e.g. poly(bis(trimethylsilyloxy)methylsilyl methacrylate) (PMATM2). The degradation kinetics of such hydrolyzable silicone coatings were assessed by mass loss measurements during immersion in deionized water. Coatings containing PMATM2 exhibited a maximum mass loss after 12 weeks, whereas PCL-based coatings showed no significant mass loss after 24 weeks. Dynamic contact angle measurements revealed the modifications of the coatings surface chemistry with an amphiphilic behavior after water exposure. The attachment of macrofoulers on these coatings were evaluated by field tests in the Mediterranean Sea, demonstrating the short or long-term antifouling effect of these hydrolyzable polymers embedded in the silicone matrix. The settlement of *A. amphitrite* barnacles on the different coatings indicated inhospitable behaviors towards larval barnacles for coatings with at least 15 wt % of additives.

## 1. Introduction

The colonization of marine species such as algae (soft fouling) and barnacles (hard fouling) on ship hulls or any other artificial submerged surfaces causes serious impacts on the marine environment [1,2]. It is noteworthy that such biofouling colonization on ship hulls increases their hydrodynamic drag leading to an increase in fuel consumption and greenhouse gases emission [3,4]. Following the international ban of tributyltin, almost 20 years ago, innovations in the field of antifouling coatings dealt with the development of coatings based on copper and co-biocides, targeting the largest biodiversity of micro- and macro-organisms as possible. During the past 10 years, more environmentally friendly antifouling coatings were developed, mainly by reducing the biocides content. The use of biocides remains the most efficient way to prevent the settlement of marine species [5,6,7,8,9], however the regulations around the world tend to drastically reduce over the years the number of harmful chemical substances authorized in antifouling paints (e.g. the European Biocidal Product Regulation) [7,10].

Fouling Release Coatings (FRC) are antifouling coatings mainly based on a crosslinked poly(dimethylsiloxane) (PDMS) matrix. Contrary to the commonly used antifouling coatings, they do not release biocides and their antifouling performances are based on their specific properties (a low surface free energy, a low elastic modulus and a low surface roughness), which enable the release of fouling organisms in presence of hydrodynamic forces (e.g. during navigation) [6,7,11]. Thus, their antifouling performances are strongly dependent on the ship velocity, and they tend to be fouled during idle periods. Moreover, FRC suffer from a low fouling release efficacy towards some microalgae such as *Amphora* and *Navicula* diatoms [12,13,14]. To overcome these issues, many researchers are now focusing their work on the synthesis of novel additives or polymer matrixes for developing a new generation of FRC. A promising antifouling mechanism is to create inhospitable surfaces which weakens the strength of adhesion between the bioadhesive, secreted by the fouling organisms, and the submerged surface. Inhospitable surfaces can be obtained using surface-modifying compounds, such as blended or grafted amphiphilic copolymers [15,16,17,18,19,20,21], zwitterionic polymers [22,23,24,25], or hydrogel-like polymers [26,27]. Specialty additives like fillers (sepiolite nanofibers, modified graphite, carbon nanotubes) and pigments (TiO_2_, ZnO) are also known to enhance the fouling release properties once embedded in a PDMS matrix [28,29,30,31]. Besides, silicone oils used as additives in PDMS elastomers can also enhance the fouling release properties of FRC due to their appropriate critical surface tension, and to a leaching phenomenon, which leads to the detachment of fouling organisms by slipping [32,33,34,35].

In this work, three hydrolyzable polymers were embedded in a silicone elastomer to combine the hydrophobic character of PDMS with the hydrophilicity of the hydrolyzed polymer additives. With these new silicone coatings, it is expected that in contact with water, the additive polymers will migrate more or less quickly towards the upper surface and create some hydrophilic and/or hydrolyzable regions among the hydrophobic PDMS phase. Thus, these coatings are expected to behave as evolving surfaces which could prevent the settlement of fouling organisms during static periods or limit their adhesion, without the use of biocides. Hydrolyzable polymers have already been used in the field of antifouling coatings, for the development of biocides-based erodible coatings. Bressy’s group synthesized poly(trialkylsilyl methacrylate)s to develop coatings with tunable erosion profiles [36,37,38], Zhang’s and Réhel’s groups worked on degradable polyesters for antifouling applications [39,40,41,42,43,44,45,46]. Copolymers based on poly(ε-caprolactone), poly(δ-valerolactone), polybutylene succinate, poly(L-lactide), poly(ethylene adipate), and poly(2-methylene-1,3-dioxepane) were synthesized to promote the erosion of the polymer matrix and the controlled release of biocides.

In our study, three hydrolyzable polymers, with different kinetics of hydrolysis, were evaluated as polymer additives embedded in a silicone matrix. The aim was to select the most appropriate hydrolysis kinetics to improve the antifouling performances of PDMS-based coatings in static conditions. Poly(ε-caprolactone) (PCL) was chosen for its slow kinetics of hydrolysis which could lead to a more durable antifouling activity. PCL-*b*-PDMS-*b*-PCL, a triblock copolymer containing two PCL blocks and a central PDMS block, was also evaluated as its triblock structure should improve the miscibility of PCL segments in the silicone elastomer. The third hydrolyzable polymer, poly(bis(trimethylsilyloxy)methylsilyl methacrylate) (PMATM2), was selected due to its fast hydrolysis kinetics in opposition to the two previous ones [38].

The coatings developed in this study were free of fillers and pigments in order to evaluate only the influence of the polymer additives on the properties of the PDMS elastomer matrix. The silicone-based coatings with hydrolyzable polymer additives were characterized in terms of surface free energy, wetting properties, roughness, elastic modulus, and hardness, since these parameters are the main drivers of antifouling and fouling release performances of PDMS elastomer-based coatings. A mass loss test was performed to estimate the degradation kinetics of the hydrolyzable polymers embedded in the silicone matrix. The characteristics of the coatings were correlated with their antifouling efficacy during immersion in the Mediterranean Sea (Toulon bay), in static conditions for 8 months. Finally, settlement bioassays using cyprids of *Amphibalanus amphitrite* gave further information on the antiadhesive properties of these silicone-based coatings with hydrolyzable polymer additives.

## 2. Materials and Methods

### 2.1. Materials

Poly(ε-caprolactone)-diol (PCL) (*M*_n,NMR_ = 3,000 g/mol) was supplied by Perstrop (Shanghai, China) and used without further purification. The polyester modified polysiloxane triblock copolymer PCL-*b*-PDMS-*b*-PCL (with *M*_n,NMR_ = 6,800 g/mol; *M*_n,NMR_ (PDMS block) ≈ 2,200 g/mol; total PCL end blocks ≈ 4,600 g/mol), and a polydiethoxysiloxane (PDES) crosslinking agent (*M*_n,NMR_ = 744 g/mol) were kindly supplied by Evonik (Essen, Germany). Dioctyltin dilaurate (DOTDL) catalyst was kindly supplied by TIB Chemicals (Mannheim, Germany). Bis-silanol polydimethylsiloxane oil (*M*_w_ = 24,000 g/mol) was kindly supplied by Dow Corning (Seneffe, Belgium). MATM2 monomer was kindly supplied by PPG (Amsterdam, The Netherlands) and was distilled under reduced pressure in presence of 2,6-di-tert-butyl-4-methylphenol before use. AIBN (2,2’-Azobis(2-methylpropionitrile)) was purchased from Sigma Aldrich (Saint-Quentin-Fallavier, France) and recrystallized in methanol before use. CPDB (2-Cyano-2-propyl dithiobenzoate) was purchased by Strem Chemicals (Bischheim, France) and used as received. Xylene reagent was purchased from Carlo Erba Reagents (Peypin, France) and distilled under reduced pressure before use. Diiodomethane (CH_2_I_2_, Sigma-Aldrich, Saint-Quentin-Fallavier, France) was used as received.

### 2.2. Synthesis of PMATM2

The PMATM2 homopolymer was prepared by a Reversible Addition-Fragmentation chain Transfer (RAFT) polymerization in xylene with AIBN as an initiator and CPDB as the transfer agent, as previously described by Lejars et al. [38]. The reaction mixture was stirred at 75 °C for 20 h under nitrogen atmosphere. The synthesized polymer was precipitated in a great excess of cold methanol to provide a very sticky pink polymer. The reaction was followed by ^1^H NMR analysis to assess the conversion of MATM2 monomer by measuring the disappearance of the vinylic protons at 5.8 ppm [38]. Once the conversion no longer evolved, the reaction was stopped. The number average molar mass (*M*_n, NMR_) of the purified PMATM2 was assessed by ^1^H NMR analysis assuming that there is one CPDB molecule per polymer chain using Equation (1).

^1^H NMR (PMATM2) (400 MHz, CDCl_3_, δ, ppm): 7.9 (2H, m, aromatic –CH–), 7.5 (1H, m, aromatic –CH–), 7.3 (2H, m, aromatic –CH–), 1.7-2.5 (2H, wide signal, –CH_2_–), 0.8-1.5 (3H, wide signal, –CH_3_), 0.2–0.3 (3H, s, –SiCH_3_), 0.02-0.2 (18H, m, (–[SiCH_3_]_3_)_2_).
(1)$$M_{n,NMR}\left(g/mol\right)=\frac{I_{0.02-0.3}^{SiCH3}/21}{I_{7.9}^{CHaromatic}/2}\times M_{repeatunit}+M_{CPDB}$$

$I_{0.02-0.3}^{SiCH3}$ corresponds to the intensity of the signal at (0.3–0.02) ppm (21H, m, –SiCH_3_). $I_{7.9}^{CHaromatic}$ corresponds to the intensity of the signal at 7.9 ppm (2H, m, aromatic –CH–) corresponding to the CPDB moiety. *M*_repeat unit_ corresponds to the molar mass of the repeat unit (305 g/mol). *M_CPDB_* corresponds to the molar mass of the RAFT agent moiety (221.34 g/mol).

### 2.3. Characterization Techniques of the Polymer Additives

#### 2.3.1. Nuclear Magnetic Resonance Spectroscopy (NMR)

^1^H NMR spectra were recorded on a Brüker Advance 400 (400 MHz) spectrometer, in deuterium chloroform (CDCl_3_), at room temperature.

#### 2.3.2. Size Exclusion Chromatography (SEC)

The number average molar mass (*M*_n,TD-SEC_) and dispersity (Đ_M_,TD-SEC) were measured on a Triple Detection Size Exclusion Chromatography (TD-SEC) with OmniSEC software (Viscotek, Vénissieux, France). The instrument is composed of a GPC Max (comprising a degazer, a pump and an autosampler) with TDA-302 (RI detector, right and low angle light scattering detector at 670 nm and viscometer) and UV detector (Knauer, Berlin, Germany). The following Viscotek columns were used: a HHR-H precolumn, a GMHHR-H and a GMHHR-L ViscoGel columns. THF was used as the eluent with a flow rate of 1.0 mL/min at 30 °C. Purified polymers were dissolved in THF at ca. 10 mg/mL and filtered on a 0.2 µm PTFE filter. Molar masses were calculated using the value of the refractive index increment d*n*/d*c* for PMATM2 (d*n*/d*c*= 0.044 mL/g in THF) and PCL (d*n*/d*c*= 0.071 mL/g in THF) using the OmniSEC software [38,47]. The analysis of the PCL-*b*-PDMS-*b*-PCL triblock copolymer was not possible by TD-SEC in THF due to its refractive index similar to *n*_THF_, resulting in no signal with the RI and light scattering detectors. This triblock copolymer was thus analyzed on a SEC apparatus (Waters, Milford, CT, USA) equipped with a refractive index detector, with toluene as eluent. The columns used were Styragel HR1, HR4, and HR5. Toluene was used as the eluent at a flow rate of 1.0 mL/min. Polystyrene standards (*M*_n_ values varying from 400 to 20,000 g/mol) were used to generate a conventional calibration curve. Data were analyzed using Breeze software (Waters, Milford, CT, USA).

#### 2.3.3. Differential Scanning Calorimetry (DSC)

The thermal behavior of polymers was analyzed on a Differential Scanning Calorimetry equipment (TA Instruments, New Castle, DE, USA) under a nitrogen flow of 50 mL/min. To determine the crystallinity (X_c_) of PCL and PCL-*b*-PDMS-*b*-PCL, samples (10–15 mg) were first heated at 100 °C for 2 min, then cooled down to −90 °C at a rate of 1 °C/min, kept at −90 °C for 2 min, and then reheated to 100 °C at a rate of 1 °C/min. All values were recorded from the second heating run. To determine the glass transition temperature of the amorphous PMATM2, the sample was cooled down to −90 °C at a rate of 20 °C/min, kept at −90 °C for 2 min, and then reheated to 30 °C at a rate of 20 °C/min. To determine the glass transition temperature of the semi-crystalline PCL, modulated DSC was performed, the sample was first heated at 100 °C for 2 min then cooled down to −90 °C at a rate of 5 °C/min, kept at −90 °C for 2 min, and then reheated to 100 °C at a rate of 2 °C/min (conventional DSC experiment did not allow the visualization of the glass transition, regardless of the program choice). 

### 2.4. Preparation of Polymer Additive Films

To assess the wettability properties of the polymer additives alone, each polymer was dissolved in chloroform (50 wt %) and consecutively applied with a bar coater (100 µm wet) on a thin sandblasted poly(vinylchloride) (PVC) substrate (2 × 4 cm^2^) previously cleaned with ethanol. The samples were dried for 48 h at room temperature before contact angles analysis.

To determine the mass loss kinetics of the three hydrolyzable polymers, the chloroform-based solutions were also casted on glass slides (76 × 26 mm^2^) previously cleaned with acetone. They were immersed in deionized water for a certain amount of time until the polymer films onto the glass slides cracked into pieces and could no longer be weighed. 

### 2.5. Preparation of PDMS-Based Coatings

The hydrolyzable additives-based silicone coatings were prepared as described in Figure 1. The hydrolyzable polymers were dissolved in xylene (60–80 wt %) for 15 min of stirring. The xylene-based solutions were consecutively dispersed within the bis silanol-terminated PDMS at 1500 rpm for 20 min with a Dispermat® apparatus. The DOTDL catalyst was added using a micropipette). The PDES crosslinking agent was further added into the solution at 1500 rpm for 10 min. The massic ratio of PDMS:PDES:DOTDL was 100 : 4.4 : 0.1 for all the mixtures. The solvent amount (21–26)%was adjusted to have a similar viscosity for all formulations (full description in Table 1). The coatings were prepared immediately after mixing (to avoid phase separation), by casting the formulation onto various substrates. The coatings were cured at room temperature for 24 h with a relative humidity of 38%–48%, measured by a hygrometer. For contact angle measurements, roughness measurements and the hydrolytic degradation tests, samples were prepared by casting ca. 2 g of the formulations onto cleaned glass microscope slides (76 × 24 mm^2^). Samples for dynamic mechanical analyses (DMA) investigations were 1 mm-thick free standing films obtained after casting formulations onto 5 × 5 cm^2^ cleaned smooth PVC panels, and gently detached from the substrate with a plastic tweezer. The samples for DMA analyses were also used for the hardness measurements by stacking 6 specimens. Additional applications over 10 × 10 cm^2^ sandblasted and cleaned PVC panels were performed with a bar coater (wet thickness of 300 µm) for field tests and biological assays. The nomenclature of the coatings is YYY-BX, where YYY is the type of hydrolyzable polymer additives (PCL for the PCL homopolymer, TGO for the triblock PCL-*b*-PDMS-*b*-PCL and M3T for the PMATM2 homopolymer), B stands for blend, and X corresponds to the mass fraction of the hydrolyzable polymer within the dry coating.

Due to the limited miscibility of PCL within the silicone matrix, the maximum amount of PCL was up to 15 wt %. Above this quantity, the coatings with PCL-based additives displayed a macroscopic phase segregation, while coatings with 20 wt % of PMATM2 could be achieved without incompatibility issues at the macroscale level.

### 2.6. Hydrolytic Degradation of Coatings

The mass loss of hydrolyzable additives-based silicone coatings was carried out by immersing coated glass slides for 24 weeks in deionized water. Before gravimetric measurements, the coatings were gently rinsed with deionized water and dried at room temperature for 6 h. The reported mass loss results are the mean value of three replicates, weighted on an analytic balance (Denver Instrument) with a precision of 0.1 mg and calibrated prior to each measurement. The mass loss (wt %) was calculated using Equation (2).
(2)$$Mass\;loss\;\%=\frac{\left(Wo-Wt\right)}{\left(Wo-Ws\right)}\times 100$$
where *W*o is the initial mass of the coated glass slide (g), *W*t is the mass of the coated glass slide after an immersion time *t* (g) and Ws is the mass of the non-coated glass slide (g).

### 2.7. Contact Angle Measurement and Surface Free Energy of Coatings

Static contact angle measurements were performed using a DSA 30 apparatus (Krüss, Hambourg, Germany) by the sessile drop technique under ambient conditions. Five contact angle measurements were carried out with 2 µL-droplets of deionized water (θ_w_) and diiodomethane (θ_diiodo_), after 4 s of stabilization. The surface free energy of the coatings ($\gamma_{S}$) was determined by the Krüss Advance software using the Owens Wendt method [6]. Both the dispersive ($\gamma_{S}^{D}$) and the polar ($\gamma_{S}^{P}$) components of the surface free energy were assessed ($\gamma_{S}=\gamma_{S}^{D}+\gamma_{S}^{P}$). Measurements were performed on pristine samples and samples immersed in deionized water for 5 weeks.

### 2.8. Dynamic Contact Angle Measurements of Coatings (DCA)

DCA experiments were carried out by the advancing-receding drop method using the DSA 30 apparatus (Krüss, Hambourg, Germany) under ambient conditions. A 4 µL-deionized water drop was first placed onto the coating with the syringe tip still immersed within the droplet, than the droplet is grown at a rate of 0.75 µL/s until a final volume of 25 µL for the measurement of the advancing contact angle (θ_w,adv_). The receding contact angle (θ_w,rec_) was measured by withdrawing the liquid at the same rate. For each coating, the reported θ_w,adv_ and θ_w,rec_ were the average values obtained from 1 cycle of advancing-receding on 3 deionized water droplets.

### 2.9. Surface Roughness Measurements of Coatings

Surface roughness profiles were measured by a contact type stylus profiler (Taylor Hobson, Leicester, UK) using a 2 µm radius tip and a 0.1 µm radius diamond tip, with a minimum applicable 1 mN stylus load. The stylus moved across 15 mm length of the coating surface, at a constant velocity of 0.50 mm/s to obtain surface height variations. R_a_ values were assessed from the average of three measurements. According to ISO 4288-1996, the selected cut-off wavelength (low-pass filter) was λ_c_ = 0.8mm since R_a_ < 2 µm. The high-pass filter (λ_s_) was fixed at 0.0025 mm for all the samples.

### 2.10. Hardness Measurements

The hardness of the coatings were determined with a Shore A Durometer (Hildebrand, Oberboihingen, Germany) which indicates the relative resistance of a material to indentation with a load applied to the indenter. The reported hardness values were the average of 5 measurements on 6 mm-thick stacked specimens after 15 s of stabilization according to the standard ISO 868.

### 2.11. Dynamic Mechanical Analysis of Coatings

The dynamic mechanical analyses were performed using free standing films of 10–14 mm length, 8–10 mm width and 0.6–1.0 mm thick. The storage modulus E’ of samples was determined at 25 °C with a dynamic mechanical analyzer DMA (TA Instrument Q800, New Castle, DE, USA) operating in tensile strain-sweep mode. A frequency of 1 Hz, a preload of 0.01 N, and amplitudes from 5 to 50 µm (within the linear viscoelastic region) were used. The results are the average of at least 5 measurements on 3 different samples. 

### 2.12. Marine Field Test 

The coated panels (10 × 10 cm^2^) were fully immersed in a vertical position in the Mediterranean Sea, in the Toulon Bay (43°06′25″N; 5°55′41″E), at a relatively shallow depth of water (1 m). The Antifouling (AF) and Fouling Release (FR) performances in static conditions were evaluated every month, over a period of 8 months (from June 2017 to January 2018). Two replicates of each coating were investigated, from which one was partially-cleaned with a sponge, each month, for assessing the FR properties. A sand-blasted uncoated PVC panel was also immersed as a negative control. A French practice adapted from the French NF T 34-552 standard was used to assess the AF efficacy of the coatings. This standard requires to report: (i) the type of macrofoulers attached to the surface, and (ii) the estimated percentage of the surface covered by each type of macrofoulers (intensity factor, i.e. IF, see Table 2). The inspection was performed 1 cm from the edges of the panel. An efficacy factor N was defined as follows in Equation (3).
(3)N=∑(IF×SF)
where SF is defined as a severity factor (see Table 3), which takes into account the frictional drag penalty of ship hulls attributable to increased surface roughness due to foulers [48,49]. The best antifouling efficacy is assigned to coatings with a N value of 5, which corresponds to a surface fully covered by a biofilm. The worse antifouling efficacy is attributed to the negative control with a N value which tends to ca. 40.

The FR performance, ranging from 0 to 2 (0 = worst, 2 = best), was evaluated by cleaning the coating with a sponge on the lower half of the coated panel and assessing the level of detachment of the marine organisms.

These two inspection procedures provide meaningful information on both the AF/FR activity of the coatings and their long-term durability in an aggressive, real-world environment. However static immersion tests may be susceptible to the seasonal diversity and abundance of fouling, at the test site [50]. The conclusions on the field test will thus consider the seasonal fouling activity (provided in Appendix A).

### 2.13. Larval Barnacle Culture and Settlement Assay of A. Amphitrite Cyprids

Adult and larval barnacle cultures were performed with *Amphibalanus* (=*Balanus*) *amphitrite* as described by Othmani et al. [51]. For anti-settlement assays, 2 mL of filtered sea water (FSW) were pipetted onto four different locations of each coating. Active cyprids were pipetted from the stock solution (~100 cyprids/mL) together with 200 μL of FSW. Thus, around 10 to 20 cyprids were allocated into each drop while adding 200 μL to the 2 mL drop of FSW. Incubation of cyprids on coating samples was carried out, at 22 °C, for 7 days, in a humid chamber to avoid excessive evaporation of water in the dark. This procedure allowed efficient settlement and transformation of cyprids, after which, time attached or metamorphosed individuals were counted under a binocular microscope. A poly(styrene) (PS) plate was used as control to evaluate the ability of the larvae to settle. A one-way ANOVA (Analysis of Variance) followed by post-hoc tests (Tuckey) was performed to determine which of the coatings showed a significant difference compared to the control for cyprid adhesion. The adhesion percentage is defined as the number of alive fixed cyprids divided by the total number of tested cyprids. Barnacle larvae are prone to settle on the inert PS control surface with adhesion of 80%–100% of the larvae. Further details of the bioassay are supplied in Appendix A.

## 3. Results and discussion

### 3.1. Physicochemical and Thermal Properties of the Polymer Additives

PCL, PCL-*b*-PDMS-*b*-PCL and PMATM2 were characterized by ^1^H NMR and SEC to assess their number-average molar mass as well as their dispersity (Ɖ_M_), as shown in Table 4. Contrary to PCL and PMATM2, the determination of *M*_n,SEC_ and Ɖ_M_ for the PCL-*b*-PDMS-*b*-PCL triblock copolymer was not possible on the TD-SEC equipment since the polymer has a similar refractive index as the eluent (THF) resulting in an absence of RI and LS detectors signals. Thus, PCL-*b*-PDMS-*b*-PCL was analyzed on a SEC apparatus with toluene as eluent (using a conventional calibration with polystyrene standards).

DSC was used to assess the degree of crystallinity, their melting temperature (*T*_m_) when existing, and the glass transition temperature (*T*_g_) of the three polymers.

The polymer additives displayed molar masses below 10,000 g/mol. The PCL-*b*-PDMS-*b*-PCL triblock copolymer with blocks of respective molar masses 2,300-*b*-2,200-*b*-2,300 exhibited a lower crystallinity (28%) than a PCL of 4,000 g/mol (*X*_c_ = 77%) and a PCL of 2,000 g/mol (*X*_c_ = 60%) due to its central PDMS block which brings flexibility to the polymer chain [52]. PCL and PCL-*b*-PDMS-*b*-PCL are semi-crystalline polymers with *T*_m_ values around 50–55 °C as expected, while PMATM2 is totally amorphous [52,53]. At room temperature, this methacrylic polymer is very sticky, and has a *T*_g_ value at −20 °C. These three polymer additives were chosen due to their different crystallinity degrees which may influence the final properties of coatings. 

It is worth noting that the *T*_g_ of the PDMS block for PCL-*b*-PDMS-*b*-PCL was not determined due to the inability of the cooling system to reach −140 °C (*T*_g_ of PDMS is expected to be ca. −127 °C) [53]. The *T*_g_ of the PCL block for PCL-*b*-PDMS-*b*-PCL was also not visible on the modulated DSC thermogram and was not further investigated.

### 3.2. Wettability Properties and Hydrolysis Kinetics of the Polymer Additives Films

The wettability and hydrolysis kinetics of the three polymer additives were investigated (Table 5). The three polymers are hydrophobic with θ_w,adv_ higher than 100° similar to the PDMS reference. The most hydrophobic polymers were PMATM2 and PCL-*b*-PDMS-*b*-PCL with θ_w,adv_ around 115°. This result is explained by the presence in both polymers of siloxane functions. The receding contact angle highlights the contribution of the hydrophilic phases of the coating surface [55]. According to Table 5, the three hydrolyzable polymers showed θ_w,rec_ around 50°. This contact angle hysteresis (θ_w,adv_–θ_w,rec_) reveals a chemical surface reorganization, with a reorientation towards the liquid of the hydrophilic functions of the polymers. This information is important for the following DCA measurements for PDMS-based coatings. Indeed, thanks to the receding contact angle, it will be possible to know if the polymer additive migrates and/or is present on the surface, by comparison with the PDMS reference. Given that PCL and PCL-*b*-PDMS-*b*-PCL coatings cracked into small pieces after only 24 h in deionized water, the values of hydrolysis kinetics in Table 5 are taken from the literature. According to Azemar et al., PCL (3,000 g/mol) and PCL-*b*-PDMS-*b*-PCL (≈ 2,850-*b*-2,300-*b*-2,850) hydrolyzed very slowly, ≈ 2 wt% and ≈ 17 wt% of M_n,SEC_ loss after 50 days in deionized water [54]. PMATM2 exhibited 30 wt% of mass loss after only 9 days. After this time, the PMATM2 coatings entirely cracked into small pieces as it became brittle.

### 3.3. Hydrolytic Degradation of PDMS-Based Coatings

The aim of the hydrolytic degradation test was to assess the degradation kinetics of the hydrolyzable polymers embedded in the silicone elastomer matrix when immersed in deionized water. Aliphatic polyesters such as PCL are expected to degrade very slowly through a bulk erosion [54], whereas PMATM2, bearing silylated ester groups as hydrolyzable pendant groups, is known to quickly hydrolyze (Table 5) [38]. The predominance of the PDMS matrix within our coatings may slow down the hydrolytic degradation of the polymer additives, due to the hydrophobic nature. Nevertheless the well-known mobility of siloxane bonds does not prevent the migration or diffusion of small polymers towards the surface [56]. Figure 2 shows that the PDMS elastomer coatings comprising PCL or PCL-*b*-PDMS-*b*-PCL exhibited a low mass loss value around 0.48 ± 0.08 wt % which is comparable to the additive-free PDMS reference mass loss during 24 weeks of immersion (Appendix A) [57]. The coatings comprising PMATM2 showed a maximum mass loss reached after 12 weeks. M3T-B5, M3T-B10, M3T-B15, and M3T-B20 showed mass losses of 3.4 ± 0.1 wt %, 7.8 ± 0.2 wt %, 13.3 ± 0.9 wt %, and 12.4 ± 0.3 wt % respectively, after 24 weeks of immersion. The mass loss mostly corresponds to the weight fraction of the pendant ester groups released from the blend once hydrolyzed (74 wt % of the repeating MATM2 unit is released after hydrolysis cleavage). Thus, this result indicates that poly(methacrylic acid) (PMAA) chains coming from the PMATM2 hydrolysis remained within the elastomer coatings after 24 weeks of immersion. With this mass loss test, two different kinetics profiles of hydrolysis have been observed for PMATM2 and the PCL-based additives embedded in a silicone matrix.

### 3.4. Physicochemical Properties of PDMS-Based Coatings

The surface and viscoelastic properties of the coatings were investigated to evaluate the influence of the hydrolyzable polymer additives on the specific properties of silicone-based coatings. The physicochemical and mechanical parameters are known to be directly related to the antifouling and fouling release performance of FRC. Table 6 indicates that all coatings were hydrophobic (θ_w_ ≥ 100°) with a low surface free energy ($\gamma_{S}$ ≈ 15–22 mJ/m^2^), and a low roughness (R_a_ ≈ 0.5 µm) which are in favor of reducing hydrodynamic drag [6]. The presence of PCL segments on the surface may explain why some values of roughness were slightly higher at ca. 1.5 µm. This is likely due to a phase segregation between PDMS and PCL segments (when PCL content is higher than 10 wt %). Low hardness Shore A values (15–25) and low tensile elastic moduli (E’ ≤ 2 MPa) from DMA investigations confirmed that the hydrolyzable additive-based silicone elastomer coatings exhibited similar bulk mechanical properties than the PDMS reference. This was an important point, as the higher surface mobility in PDMS elastomers allows the bioadhesive to slip during interfacial failure [58,59,60,61], which strongly influence the fouling release ability of FRC. These characterizations showed that the various crystallinity degrees of the three additives as well as their amount in the silicone matrix did not significantly influence the surface and mechanical properties of PDMS-based coatings.

### 3.5. Dynamic Contact Angle before and after Immersion

The aim of dynamic contact angle analysis was to assess whether the hydrolyzable polymer additives are located at the surface rather than trapped in the coating bulk, and if there is a significant reorganization of the surface of the coatings after contact with water. In the case of PCL-based coatings this information is relevant since the mass loss test was not sufficient to assess if polyester segments are present or not at the surface. PDMS surfaces are known to be hydrophobic and very resistant to surface reorganization in contact with water, resulting in a low contact angle hysteresis (Δθ_w_ = θ_w,adv_ − θ_w,rec_), i.e. the advancing water contact angle (θ_w,adv_) and the receding water contact angle (θ_w,rec_) are broadly similar [62,63]. Contrarily a contact angle hysteresis higher than 30° reveals a dynamic surface capable of a significant reorganization mostly due to the presence of surface active groups [64].

Figure 3 shows that all advancing contact angles were around 105°–110° indicating that all the coatings were hydrophobic (as already shown in Section 3.4). θ_w,adv_ of M3T-BX were even up to 115° probably due to the trimethylsilyl groups of PMATM2 which amplify the coating hydrophobicity. The receding contact angles highlight the contribution of the hydrophilic phases of the coating surface [55]. As expected, the PDMS reference showed a θ_w,rec_ of 98.4° ± 0.7° which means there was no significant surface reorganization. PCL-BX and TGO-BX had θ_w,rec_ between 45°–70° and 65°–85° respectively. These values indicate that PCL segments were present on the surface (see the θ_w,rec_ of PCL and PCL-*b*-PDMS-*b*-PCL previously shown in Table 5) and influenced a lot the surface wettability. The difference of θ_w,rec_ between PCL-BX and TGO-BX suggests that the PDMS block of the PCL triblock copolymer decreased the availability of PCL segments at the coatings surface. M3T-BX coatings exhibited receding contact angles even lower (θ_w,rec_ = 30°–40°) certainly due to the presence of carboxylate groups generated after the hydrolysis of the silyl ester pendant groups located at the near surface. The difference in contact angle hysteresis between PCL-based (Δθ_w_ ≈ 30°–40°) and PMATM2-based coatings (Δθ_w_ ≈ 60°–80°) may be attributed to the ability of PMATM2 to hydrolyze faster in contact with water contrary to PCL-based additives. Additionally, it is worth noting that by increasing the amount of hydrolyzable polymer additives within the silicone coating, the contact angle hysteresis was even higher.

After 3 weeks of immersion in deionized water, the advancing contact angle of hydrolyzable polymer-based coatings decreased to ca. 100° while the PDMS reference still exhibited a θ_w,adv_ of 108° (Figure 4). Thus, after 3 weeks, some hydrolyzable polymer additives may have migrated at the surface and brought further hydrophilicity to the surface. Receding contact angles were all comprised between 50° and 80° while the PDMS reference had a contact angle hysteresis of ca. 40° suggesting that the effect of surface active groups became even more apparent after 3 weeks of immersion. The increase of Δθ_w_ for the PDMS reference was unexpected, and attributed to the presence of siliceous domains coming from the self-condensation of alkoxysilane crosslinking agent as already encountered in literature [62,63,65]. Dynamic contact angles were not pursued beyond 3 weeks, due to this latter observation, which make difficult the interpretation of the results.

### 3.6. Marine Field Test

Field tests were performed to evaluate the antifouling and fouling release properties of the coatings after 2, 6, and 8 months of immersion in the Mediterranean Sea (Toulon Bay). The evolution of the N factor for all coatings during the 8 months of immersion in natural seawater as well as the coverage percentage and the type of marine organisms present on the coated panels are shown in Appendix A. After 2 and 6 months of immersion, PCL-BX and TGO-BX coatings exhibited very similar N values as the PDMS reference suggesting that the hydrolyzable polyester additive does not have any effect on the antifouling property at this stage (Figure 5). After 8 months, PCL-BX and TGO-BX coatings displayed better antifouling properties (N = 14–21) than the PDMS reference (N = 30). This result could be likely due to the increased amphiphilic nature of the coating surface after a long immersion time which make it ambiguous towards marine species [66]. M3T-BX coatings behave in a different pattern: after 2 months of immersion, N values were around 18 indicating that the fast hydrolysis kinetics of PMATM2 was very prone to discourage the adhesion of fouling organisms whatever the amount of additives. However, after 6 months, their antifouling efficacy was lost as the coatings became even more fouled (N = 30–40) than the PDMS reference (N = 30). This was attributed to the remaining PMAA chains which bring hydrophilicity on the surface making it more attractive for marine species. This short-term efficacy of M3T-BX coatings can be correlated to the mass loss test where the hydrolytic degradation had reached a plateau after 12 weeks.

Table 7 shows that the Fouling Release (FR) properties of the PDMS reference during immersion was constant but the cleaning was not sufficient to remove all macrofoulers (bryozoans in particular, Appendix A). Even if M3T-BX coatings lost their antifouling efficacy after 6 months they seem to maintain their FR abilities even after 8 months. PCL-BX and TGO-BX also exhibited durable FR properties, with a better aspect than the PDMS reference coating after 8 months as shown in Appendix A.

### 3.7. Settlement Assay of Amphibalanus Amphitrite Cyprids

The aim of this assay was to investigate the influence of variations in the surface chemistry of silicone-based coatings on the settlement of hard macrofoulers such as barnacles. Indeed, given that PDMS, PCL-BX, TGO-BX and M3T-BX coatings exhibited similar elastic moduli and hardness Shore A. Thus, any difference between the coatings in terms of barnacle attachment is attributed to the different chemical compositions, achieved by the addition of hydrolyzable polymers in the PDMS elastomers. The *A. amphitrite* settlement assay adapted from Othmani et al. was relevant to determine the antiadhesive property of the coatings towards barnacle cypris larvae (Figure 6) [51]. TGO-B15 and PCL-B15 both showed a statistically significant lower adhesion of barnacle cypris larvae than the PS control (Figure 6). Thus the presence of both hydrophobic and hydrophilic phases (revealed by dynamic contact angles analyses) potentially affects their adhesion. PMATM2-based coatings, M3T-B15 and M3T-B20, also showed significant effect on adhesion, with values lower than 30% of adhered cypris larvae. This could be correlated with the mass loss recorded after 1 week of immersion given that the release of degradation products can make the surface brittle and thus disrupt the macrofoulers settlement. During the incubation time of cypris larvae (7 days), M3T-B15 and M3T-B20 displayed mass losses of ca. 2 wt %. In addition, the surface chemistry changed. Both phenomena may explain the difficulty of barnacles to maintain their adhesion on the coatings. This study demonstrated that coatings containing at least 15 wt % of hydrolyzable polymers displayed interesting anti-barnacle larvae settlement properties, as the surface chemical ambiguity tends to discourage the barnacle cypris larvae to settle.

## 4. Conclusions

We have explored the possibility to use water-hydrolyzable PCL, PCL-*b*-PDMS-*b*-PCL copolymer, and PMATM2 as polymer additives in a silicone matrix for developing AF/FR coatings. All the additive-based PDMS elastomer coatings showed a low surface free energy, a low elastic modulus as well as a surface chemical reorganization during water immersion which make them suitable for marine applications. Particularly, PMATM2-based coatings provided an interesting evolving surface chemistry thanks to a fast hydrolysis rate of the pendant ester group and a very dynamic surface chemistry reflected by a water contact angle hysteresis up to 60°. The antifouling efficacy of PMATM2-based elastomers gave slightly better results than the additive-free PDMS coating for the first 2 months of field immersion due to the fast hydrolysis of pMATM2. Nevertheless, after 6 months of field immersion the antifouling behavior of these coatings decreased which indicates that the hydrolysis did not last enough to provide durable antifouling properties.

Finally, the importance of hydrolysis kinetics has been highlighted in this article. For antifouling applications, PMATM2 is suitable for coatings with a short-term efficacy and good antiadhesive properties towards barnacle larvae. Although PCL and PCL-*b*-PDMS-*b*-PCL could not hydrolyze in the silicone elastomer after 24 weeks of water immersion, PCL-BX and TGO-BX coatings exhibited a longer antifouling efficacy in field than the additive-free PDMS coating, and they also showed good *A. amphitrite* antiadhesive properties with 15 wt % of hydrolyzable polymer content.

## Figures and Tables

**Figure 1 polymers-11-00305-f001:**
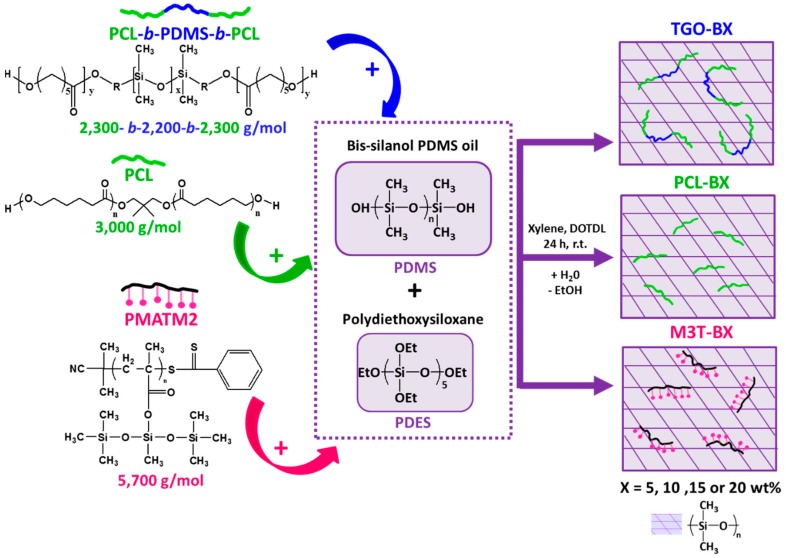
General pathway for the preparation of hydrolyzable additive-based silicone elastomers, by blending hydrolyzable polymers into a silicone condensation cure system composed of polydiethoxysiloxane, bis-silanol poly(dimethylsiloxane) (PDMS) oil and a tin catalyst. PDES: polydiethoxysiloxane; PCL: poly(ε-caprolactone); PMATM2: poly(bis(trimethylsilyloxy)methylsilyl methacrylate.

**Figure 2 polymers-11-00305-f002:**
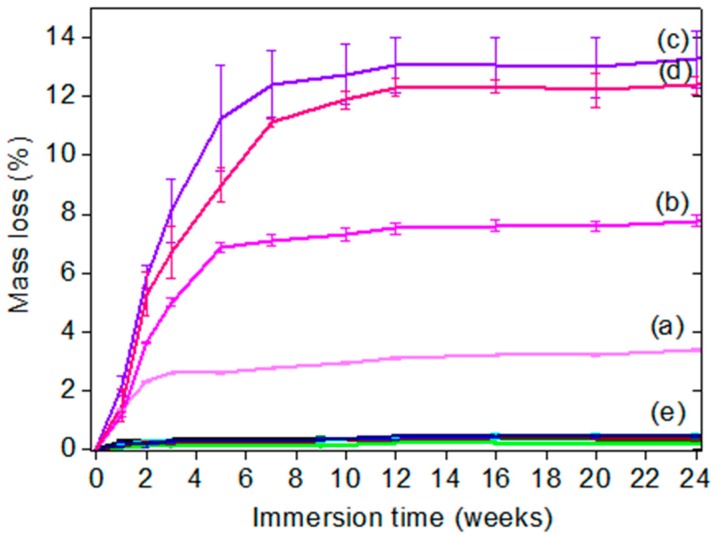
Mass loss (wt %) in deionized water at room temperature of (**a**) M3T-B5, (**b**) M3T-B10, (**c**) M3T-B15, (**d**) M3T-B20 and (**e**) the other coatings based on PCL and PCL-*b*-PDMS-*b*-PCL as well as the PDMS reference without polymer additives.

**Figure 3 polymers-11-00305-f003:**
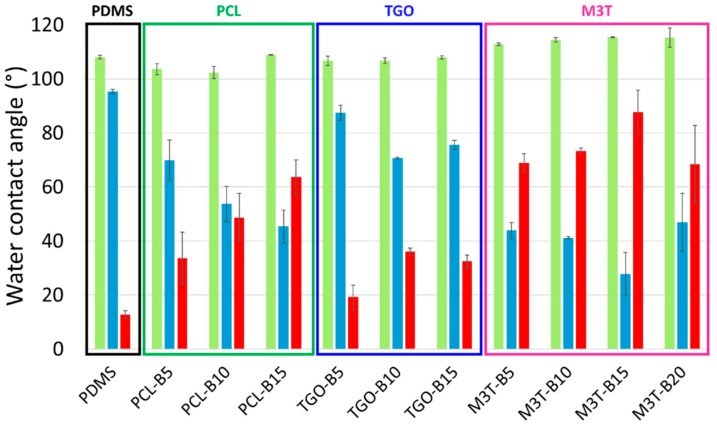
Dynamic water contact angles of the PDMS reference, PCL-BX coatings (dark green framing), TGO-BX coatings (dark blue framing) and M3T-BX coatings (pink framing) with the advancing contact angle (light green), the receding contact angle (light blue) and the contact angle hysteresis (red) before immersion.

**Figure 4 polymers-11-00305-f004:**
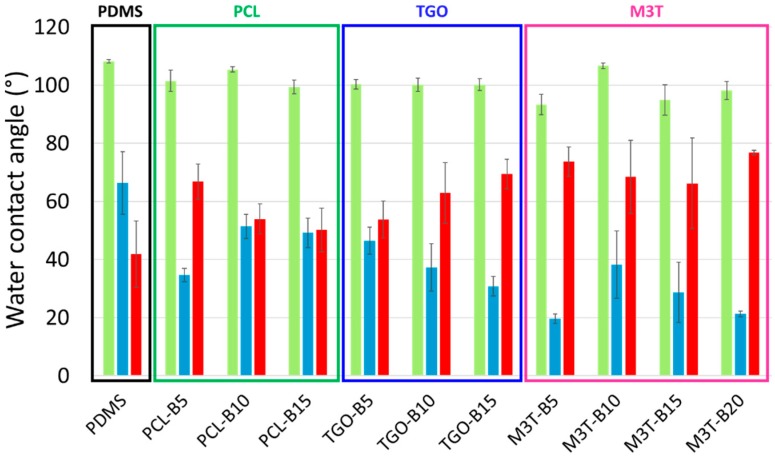
Dynamic water contact angles of the PDMS reference coating (black framing), PCL-BX coatings (dark green framing), TGO-BX coatings (dark blue framing) and M3T-BX coatings (pink framing) with the advancing contact angle (light green), the receding contact angle (light blue) and the contact angle hysteresis (red) after 3 weeks of immersion in deionized water.

**Figure 5 polymers-11-00305-f005:**
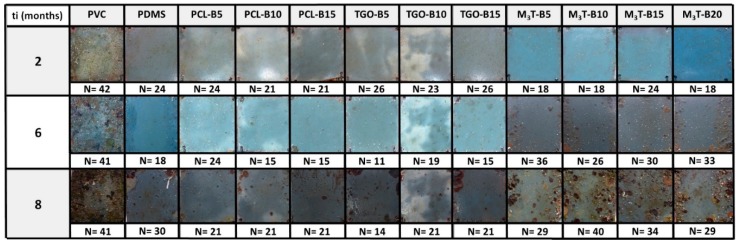
Photographs of coated panels after 2, 6, and 8 months in Toulon Bay.

**Figure 6 polymers-11-00305-f006:**
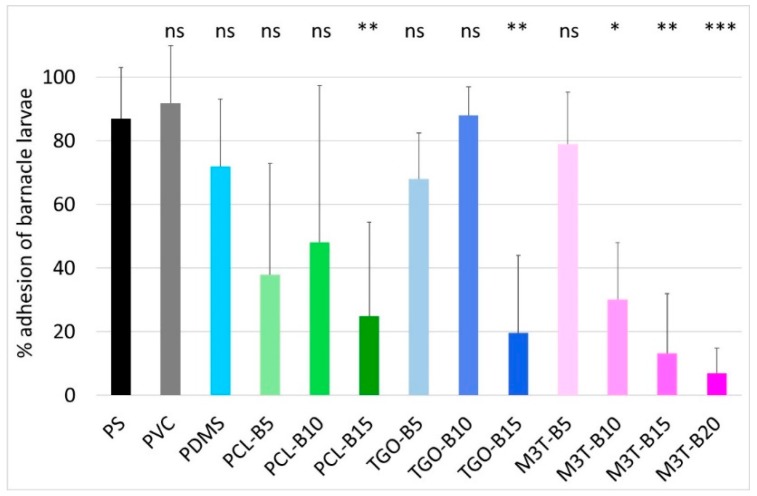
Settlement of *A. amphitrite* larvae on controls (PS, PVC) and PDMS-based coatings. Error bars represent standard deviations and asterisks indicate significant difference to the PS reference (ns = not significant; * *p* < 0.05; ** *p* < 0.01; *** *p* < 0.001).

**Table 1 polymers-11-00305-t001:** Formulation of the YYY-BX hydrolyzable additives-based silicone coatings and the PDMS reference.

Coating ID	Polymer Additive (wt %)	PDMS (wt %)	Macrocrosslinker (wt %)	Catalyst (wt %)	Solvent (wt %)
**PDMS reference**	0	74.35	3.27	0.07	22.30
**PCL-B5**	3.93	71.43	3.14	0.07	21.43
**PCL-B10**	7.68	66.18	2.91	0.07	23.16
**PCL-B15**	11.32	61.37	2.70	0.06	24.55
**TGO-B5**	3.93	71.43	3.14	0.07	21.43
**TGO-B10**	7.68	66.18	2.91	0.07	23.16
**TGO-B15**	11.32	61.37	2.70	0.06	24.55
**M3T-B5**	3.93	71.43	3.14	0.07	21.43
**M3T-B10**	7.68	66.18	2.91	0.07	23.16
**M3T-B15**	11.32	61.37	2.70	0.06	24.55
**M3T-B20**	14.88	56.94	2.51	0.06	25.62

**Table 2 polymers-11-00305-t002:** Evaluation of the fouling coverage.

% Coverage	Intensity Factor IF
No fouling	0
0 ≤ % ≤ 10	1
10 ≤ % ≤ 20	2
20 ≤ % ≤ 40	3
40 ≤ % ≤ 60	4
60 ≤ % ≤ 100	5

**Table 3 polymers-11-00305-t003:** Evaluation of the type of fouling.

Fouling Type	Severity Factor SF
Biofilm	1
Algae (brown, red, green)	3
Non-encrusting species (hydrozoa, sponges, ascidians …)	4
Encrusting species (barnacles, tubeworms, spirorbis, bryozoans, shells…).	6

**Table 4 polymers-11-00305-t004:** Characterization data of the polymer additives.

	PCL	PCL-*b*-PDMS-*b*-PCL	PMATM2
*M*_n,NMR_ (g/mol)	3000	6800	5700
*M*_n,SEC_ (g/mol)	2800 ^b^	7300 ^a^	5500 ^b^
Ɖ_M_	1.28 ^b^	1.32 ^a^	1.26 ^b^
*T*_g_ (°C)	−60 ^c^	n.d. ^d^	−20 ^e^
*T*_m_(PCL) (°C)	51	52–56	-
Δ*H*_m,sample_ (PCL) (J/g)	87	57	-
*X*_c_(PCL)(%) ^f^	62	28	0

^a^ Determined by SEC (toluene, PS standards); ^b^ Determined by TD-SEC (THF) with d*n*/d*c*= 0.044 mL/g in THF for PMATM2 and d*n*/d*c* = 0.071 mL/g in THF for PCL [38,47]; ^c^ Determined by modulated DSC; ^d^ Not determined for both blocks; ^e^ Determined by conventional DSC; ^f^ Degree of crystallinity $Xc\left(\%\right)=\frac{w\left(PCL\right)\times \Delta H_{m}\times 100}{\Delta H_{m}^{0}}$ with $\Delta H_{m}^{0}=139.3\frac{J}{g}$ and w(PCL) = 0.68 (mass fraction of PCL) for PCL-*b*-PDMS-*b*-PCL [54].

**Table 5 polymers-11-00305-t005:** Wettability properties and hydrolysis kinetics of the polymer additives films.

	PCL	PCL-*b*-PDMS-*b*-PCL	PMATM2
θ_w,adv_ (°)	104.9 ± 2.8	115.8 ± 0.7	113.3 ± 2.3
θ_w,rec_ (°)	49.0 ± 7.1	55.4 ± 2.7	40.8 ± 10.4
Hydrolysis kinetics (wt %)	2 ^a,b^ (50 days)	17 ^a,b^ (50 days)	30 ^a^ (9 days)

^a^ In deionized water; ^b^ Taken from the literature [54].

**Table 6 polymers-11-00305-t006:** Surface and viscoelastic properties of the PDMS-based coatings.

	θ_w_ (°)	γ_s_ (mJ/m^2^)	R_a_ (µm)	Hardness Shore A	Storage Elastic Modulus (MPa)
**PDMS**	110.8 ± 1.0	18.2 ± 0.8	0.13 ± 0.1	26 ± 2	0.9 ± 0.1
**PCL-B5**	109.2 ± 0.7	17.8 ± 0.6	0.28 ± 0.2	27 ± 1	1.2 ± 0.3
**PCL-B10**	107.3 ± 1.6	17.7 ± 1.3	0.35 ± 0.1	22 ± 0	1.5 ± 0.1
**PCL-B15**	108.8 ± 0.7	16.3 ± 2.2	0.23 ± 0.1	27 ± 2	1.9 ± 0.1
**TGO-B5**	104.7 ± 0.5	18.9 ± 2.1	0.42 ± 0.1	25 ± 1	1.1 ± 0.1
**TGO-B10**	102.8 ± 2.1	20.9 ± 1.8	1.42 ± 0.3	16 ± 1	1.2 ± 0.1
**TGO-B15**	102.5 ± 1.0	20.9 ± 0.8	1.51 ± 0.2	20 ± 3	2.1 ± 0.1
**M3T-B5**	103.7 ± 1.4	22.4 ± 1.3	0.58 ± 0.1	23 ± 3	1.0 ± 0.1
**M3T-B10**	106.1 ± 0.5	17.0 ± 1.3	0.25 ± 0.1	16 ± 2	1.2 ± 0.3
**M3T-B15**	103.5 ± 0.3	17.8 ± 0.7	0.47 ± 0.2	15 ± 1	1.3 ± 0.3
**M3T-B20**	107.9 ± 0.4	15.0 ± 2.2	0.49 ± 0.2	17 ± 2	1.6 ± 0.2

**Table 7 polymers-11-00305-t007:** Fouling Release properties of the PDMS-based coatings (0–2; 0 = worst, 2 = best).

Immersion time (months)	PVC	PDMS	PCL-B5	PCL-B10	PCL-B15	TGO-B5	TGO-B10	TGO-B15	M3T-B5	M3T-B10	M3T-B15	M3T-B20
2	0	1	1	1	1	1	1	1	2	2	2	2
6	0	1	1	1	1	2	2	1	1	1	1	0
8	0	1	2	2	1	2	2	1	2	1	2	1

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
