# Peer review of "Hydrolyzable Additive-Based Silicone Elastomers: A New Approach for Antifouling Coatings"

_polymers, 2019, doi:10.3390/polym11020305_

Round 1

Reviewer 1 Report

The article is well organized and well written. Some English editing can help improve the readability of the article.

The materials and methods need to be more described in details. This part must imperatively be completed. All the protocols used must be presented in detail. For example, sample preparation, analysis conditions, instrument accuracy (balance)

Author Response

Response to Reviewer 1 Comments

Point 1: The article is well organized and well written. Some English editing can help improve the readability of the article.

Response 1: The English langage was revised, in particular sentences were shortened and rephrased to appear clearer (the most modified paragraphs are the abstract, the introduction and some sections from Results and discussion visible with the “tack changes” function.

Point 2: The materials and methods need to be more described in details. This part must imperatively be completed. All the protocols used must be presented in detail. For example, sample preparation, analysis conditions, instrument accuracy (balance)

Response 2: In materials and methods, we think we brought a more thorough description of samples preparation (mostly in sections 2.4., 2.5. and 2.9.) with more details of quantities, agitation and drying time, accuracy of the balance, substrate preparation and dimensions.

Reviewer 2 Report

Dear Editor,

This manuscript of the title Hydrolyzable Additive-Based Silicone Elastomers: A New Approach for Antifouling Coatings is

1: The MS is reported the antifouling performance of various amounts of hydrolyzable polymers incorporated within a silicone matrix.

2: The MS suggested that silicon elastomers can use for the antifouling coatings.

3: The MS is well written and illustrated paper. I would recommend it for acceptance after minor points listed above.

4: The barnacle species is Amphibalanus amphitrite not Amphibalanus Amphitrite.

In this MS, there is no toxicity assay. It is better to add toxicity assay result.

Author Response

Response to Reviewer 2 Comments

The MS reported the antifouling performance of various amounts of hydrolyzable polymers incorporated within a silicone matrix.

The MS suggested that silicon elastomers can be used for the antifouling coatings.

Point 1: The MS is well written and illustrated paper. I would recommend it for acceptance after minor points listed above.

Point 2: The barnacle species is Amphibalanus amphitrite not Amphibalanus Amphitrite.

In this MS, there is no toxicity assay. It is better to add toxicity assay result.

Response 1: The English language and style was re-evaluated and modified as much as possible, sentences were shortened and some were rephrased for an easier reading.

Response 2: The capital letters of “Amphitrite” words were all replaced by a lowercase letter.

Regarding the bioassay, it is indeed interesting to investigate the toxicity of coatings, we did not plan it at the beginning but we consider doing it for the following steps of our analyses.
